# Theorizing People of Mixed Race in the Pacific and the Atlantic

Yasuko Takezawa [1] and Stephen Small [2,*]

1   Institute for Research in the Humanities, Kyoto University, Kyoto 606-8501, Japan; takezawa.yasuko.2u@kyoto-u.jp
2   Department of African Diaspora Studies, University of California, Berkeley, CA 94720, USA
*   Correspondence: small@berkeley.edu

**Abstract:** The most extensive theoretic and empirical studies of people of mixed racial descent extant today have addressed nations across the Atlantic. This article reveals how this literature on people of mixed racial descent is limited in its claims to represent a "global model". In contrast, we argue that by juxtaposing institutional factors in the Atlantic region and Japan we can expand our understanding of people of mixed racial descent across a far wider range of social and political terrains. A consideration of Japan uncovers a fascinating combination of factors impactful in the emergence of populations of mixed origins in the Pacific region more generally. By identifying this range of variables, we believe this analysis can be instructive for scholars of race focusing on the Atlantic and can contribute to a more encompassing approach for theorizing people of mixed racial descent.

**Keywords:** mixed race; Pacific; Atlantic; Japan; gender; migration

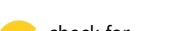



## 1. Introduction

In many nations around the world, people of mixed racial descent are growing in number and in diversity of background and are gaining more and more social acceptance and even greater visibility and attention than ever before (Daniel and Hernández 2020; Dixon and Telles 2017; Strmic-Pawl 2016). In nations across the Americas, these groups have received considerable academic attention for decades (Small 1994, 2004; Davis 1991; Heuman 1981; Jordan 1968). They continue to be studied, but in fundamentally different legal and social contexts (Root 1992, 1996; Wade 2009; DaCosta 2007; Edwards et al. 2012; Sue 2013; Telles 2014; Loveman 2014). In Europe, especially in the United Kingdom, Germany and the Netherlands, people of mixed racial descent have also received significant attention (Parker and Song 2001; Christian 2002; Campt 2004; Hondius 1999, 2014; Song 2017). Increasing attention has occurred in other regions as well (King-O'Riain et al. 2014; Rocha 2016; McGavin and Fozdar 2017; Fozdar and Perkins 2014; Williams 2017a, 2017b).

The dominant model in these studies is heavily based on transatlantic experiences, an approach that foregrounds slavery, colonialism, Black/White relations and their legacies (Joseph-Salisbury 2018; Campion 2017; Small 2004; Ifekwunigwe 2004). While this model has made significant contributions, it has several limitations, including the fact that it is based overwhelmingly on studies of Black/White people of mixed racial descent (Takezawa and Tanabe forthcoming). In addition, it is geographically limited to the Atlantic sphere and even in this sphere it no longer provides a satisfactory explanation of current demographic complexities. This is the case, for example, in studies of people of mixed race that do not involve Black/White groups, such as Asian/White or Asian/Black, which tends to focus on those living in the Atlantic. Furthermore, on both sides of the Atlantic, populations with combinations of national, racial and religious backgrounds are far more diverse than ever before, a development that has given rise to the concept of "super-diversity" (Vertovec 2007, 2019). The implications of super-diversity for the study of people of mixed racial descent are only just beginning to be explored (Fozdar and Perkins 2014).

In light of the fact that: (a) there are people of mixed racial descent in so many nations on several continents and (b) they are increasingly prominent demographically and in terms of multiple representations across various media, how do we develop a model that takes account of the experiences of people of mixed race outside of the Atlantic? In addition, how can we identify variables that shape those groups' interaction.? These are the questions that concern us.

We take Japan as one example of the unique configurations across the Pacific case, as this is a society in which biracial television personalities, newscasters, athletes, musicians and celebrities are increasingly in the limelight (Iwabuchi 2014; Takezawa 2016). The most prominent example of this group is world champion tennis player Naomi Osaka, who is of mixed Black and Japanese parentage. These groups are prominent even though only 3.3% of new marriages currently occur between Japanese nationals and non-Japanese (Ministry of Health, Labour and Welfare 2018, p. 42).[1]

In this article, we hope to contribute to expanding our understanding of people of mixed racial descent by juxtaposing the key institutional factors in the Atlantic region with the most prominent institutional factors in Japan. Needless to say, Japan does not represent the Pacific region, but consideration of Japan uncovers a fascinating combination of factors that were—and are—impactful in the emergence of populations of mixed origins in the Pacific region more generally. We believe this analysis provides key insights for scholars of race focusing on the Atlantic and expands the range of variables pertinent to an analysis of mixed racial descent beyond the Atlantic, especially as it introduces key variables (such as military bases) which are not central to analysis of the Atlantic region.

## 2. A Global Model or an Atlantic Model?

Jayne Ifekunigwe provides a useful summary of the key sets of variables deployed in the Atlantic models of analyzing people of mixed racial descent (Ifekwunigwe 2004). She identifies several pillars of "racial" mixing (Ifekwunigwe 2004, p. 6). The first are political economic/power-producing pillars, specifically European expansion, settler-colonization, and imperialism on the one hand, and Atlantic slavery on the other. The second are structural/status-defining, specifically, "Race"/color hierarchies and, separately, gender hierarchies. European expansion, settler colonialism and imperialism, including slavery, are the major forces that subjugated Black people under European rule in Africa, forcibly transported Black people to the Americas and created the political, economic and social mechanisms that led to Black people settling in Europe in unprecedented numbers in the 20th century. These factors gave rise to populations of people of mixed racial descent, first in the European colonies across the Americas (dominated by parentage of White fathers and women of color) and subsequently in Europe, where White fathers and Black mothers again played a major role. However, especially from the 20th century, the pattern changed to that of Black fathers and White mothers (Christian 2002; Campt 2004). In other words, the gender/race dynamics were different—an issue we discuss below. These variables in the research by scholars of the Atlantic exert overwhelming influence on analysis of other regions.

Closer inspection of these variables raises doubts about whether they are the most appropriate for examining other regions outside the Atlantic. For example, from the last quarter of the 20th century in Europe—and across the Americas—populations of people of mixed race have become far more diverse than the historically dominant patterns. In the United States, there are growing numbers of mixed people who do not have a Black parent descended from slavery (Strmic-Pawl 2016). In Europe, there are more people of mixed racial descent with one Black parent from Africa rather than the Caribbean, while many have parents that are Muslim, rather than Christian (Small 2018). The number of people of mixed racial descent who do not have a Black parent has also dramatically increased, including those with backgrounds in India, Pakistan, Turkey, Indonesia, Morocco and Algeria (Fresnoza-Flot 2017; Bhattacharyya et al. 2002; Hondius 2014). There are also families involving so-called mail-order brides, especially from Asian nations, the brides

of sex tourists and children from trans-racial adoption (Brennan 2004; Choy 2013). These factors do not fit neatly into the overall Atlantic model and clearly did not operate in the same way in the Pacific as they did in the Atlantic.

## 3. An Alternative Set of Variables

Given the limitations of the Atlantic model, analysis of people of mixed racial descent in other contexts needs to take account of a broader array of variables than are included in that extant model. Modifying this Atlantic approach, as summarized in Ifekwunigwe's general theory, we propose that the following six structural factors are fundamentally central and important in understanding how intergroup marriage/dating unfolded and of how people of mixed racial descent populations emerged. These variables have had a continued and sustained impact up until the present time.

### 3.1. Slavery

Slavery was the preeminent model of European colonization across the Americas, and the foundation for most populations of people of mixed racial descent. It is the invariable point of departure for pretty much every aspect of people of mixed racial descent that emerged in the Americas, and it directly led to the early patterns of interracial dating in Europe. Patterns established during slavery continued long after it was legally abolished. Yet although slavery occurred in the Pacific, it lacked any significant counterpart to patterns in the Atlantic (see Encyclopedia of World Slavery). Because of its scale and impact on the presence of mixed race in the entire trans-Atlantic region, we list this as a factor independent of #2.

### 3.2. Other Types of Conquest and Colonialism/Imperialism

This refers to racial/ethnic mixing arising from an external or internal invasion, conquest and/or colonization. External colonialism/imperialism typically takes place outside of a historical state territory due to the expansion of imperial territory and invasion of a foreign territory. This could include forced or voluntary migration, all forms of which are related to colonialism. Internal colonialism is distinguished from external colonialism, as conquest and colonization are limited within a territorial state, most typically over indigenous peoples.[2] For both patterns, mixed race/ethnic children are born to parents representing, on one side, a (former) colonizer/oppressor and on the other side, a (former) colonized/oppressed. Oftentimes, even after political colonialism ended, an aspiration for assimilation has promoted interracial mixing due to the stigma or disadvantages attached to membership in the minoritized group, as well as the implicit or explicit assimilation policy or ideology of the dominant society.

### 3.3. The Presence of Foreign Military Forces and Bases since Occupation

The occurrence of war and the establishment of foreign troops and military bases—with their range of military members and personnel (overwhelmingly male)—in various nations across Asia, especially since post World War II, led to the emergence and growth of mixed-race people. Japan (after the GHQ presence, that is the Allied occupation of Japan at the end of the second world war), South Korea, the Philippines and Guam are the most obvious cases. The dominant patterns of interracial dating and marriage in these regions, past and present, included casual sexual encounters, romantic relationships, official marriages, and tragically, rape. Since American forces included a significant proportion of African Americans, children of mixed race with Black fathers have become a highly visible presence in these societies for the first time.

### 3.4. Work Migration

This pattern of racial/ethnic mixing dates to the modern period or even earlier. There has been, at least until recently, substantial inequality in power between the flow of migration from Western countries to non-Western—most typically missionaries, teachers, senior

managers in industry and skilled laborers—and that from non-Western or non-industrial societies to industrialized ones—including political/economic refugees, students and most often unskilled or semi-skilled laborers. There are also middlemen or merchants who constantly cross borders between Western and non-Western countries. Migration patterns accelerated during globalization have generated population movements on a massive scale. Within these global patterns, there have always been migrants across nations in Asia— migrants who regarded one another as ethnically and religiously different but racially the same.

### 3.5. Marriage Migration

Transnational migration (primarily of women) specifically for the purpose of marriage with (primarily male) partners residing in a foreign country has increased. This includes historical models of so-called "mail order brides", including an increasing number of institutionally organized marriages across national boundaries, those that develop out of romantic relationships, and sometimes so-called "paper marriages", for the purpose of illegal migration (Constable 2005; Fresnoza-Flot and Ricordeau 2017).

### 3.6. Other Patterns

There are other types of structural factors that have contributed to the emergence of people with mixed heritage. Theoretically, it is not difficult to imagine the existence of children born out of one-time relationships. Examples may include children born from the rape of local women by men of different nationalities or groups during ethnic conflicts, wars, or "sex tours". There are also marriages or mixings between members of different minoritized peoples. The lack of reliable statistics prevents us from fully grasping the significance of their presence. As we mentioned in the beginning, we limit our discussion in this paper to children of mixed race/ethnicity living within a particular society. Those currently living outside of the society in question are out of the scope of this paper.

Modifying the variables in Ifekunigwe's model, we present the following four variables as significant alternative variables that intersect the six features we just presented above:

(a)  Race, including:

(a-1) Skin color and societal notions of racial categories;
(a-2) The ideology of "bloodlines";

(b)  Gender and sexuality;
(c)  Class and societal status;
(d)  Other societal and cultural differences (nationality, language, religion, etc.).

This intersectionality of variables has produced sharp differences in patterns of racial/ethnic mixing as well as representations and identifications of people of mixed heritage between the trans-Pacific and the trans-Atlantic. This is not an overarching explanation of all possible cases, however, and it is necessary to carefully trace individual examples demonstrating how the movement of the state, politics, economy and culture influence the workings of inclusion, exclusion and hierarchy in their particular temporal and geographical context.

## 4. Examples of an Alternative Model

The following are examples of our alternative model, with attention paid to the intersectionality of the six structural factors and four variables defined above. The patterns of people of mixed racial descent in the Atlantic and those in Japan (as an example of the Pacific) share some similarities, but they do not always exist in other regions or spheres due to the different types of salience and complex intersections.

### 4.1. Slavery

Slavery was the preeminent model of European colonization across the Americas, and it is under slavery that a large and significant population of people of mixed racial descent

emerged. Slavery is the invariable point of departure for more or less every aspect of people of mixed racial descent that emerged in the Americas, which directly led to the early patterns of inter-racial dating in Europe. Patterns established during slavery continued long after it was legally abolished. We omit consideration of these examples due to the abundance of the literature that discusses the intersectionality of race, gender, class and sometimes more of the mixed race of this category, as discussed above.

Slavery does not have the same applicability in the Pacific as it does in the Atlantic. Japanese historians have pointed out the existence of slavery in ancient Japan, which involved slaves engaged in digging tombs and other undesirable physical labor (Jinno 1993). The ethnic identity of these slaves remains little known, though many scholars believe they were criminals within the society and therefore ethnically the same. We believe that slavery has little or no relevance to patterns or interpretations of people of mixed race in Japan today.

### 4.2. Conquest and Colonialism/Imperialism Other than Slavery

In the last five centuries or more, no significant people of mixed racial descent have emerged in Europe resulting from invasion or conquest by populations of fundamentally different racialized groups within Europe. It is correct that Muslim populations from North Africa controlled vast parts of the Iberian Peninsula and major areas of southern France through the 1400s. Typically described in religious terms, they were also described as somehow racially different from Europe's populations. In that sense, the children that emerged could be considered racially mixed. There was also a period when Constantinople in Turkey was the center of Europe, and Turkish and both western and northern European populations intermixed, but after the 1500s, the only important conquests that took place within Europe were amongst people believed to be largely of the same racial groups—for example, the British conquering the Irish and Scottish, or the invasion by Nazis of Denmark, Netherlands, Belgium and France.

Conquest and colonialism clearly occurred in the Atlantic region—in the Americas, obviously, and across Europe from far earlier. One key difference, though, is that conquests in Europe typically involved nations that were regarded as ethnically (or religiously) distinct but not racially distinct.

In spite of the long-lasting myth of Japan as a monoethnic and a mono-racial society, during the 20th century, Japan had become distinctly multi-ethnic through its conquest and colonization, first of the Ainu and Okinawans in the 19th century, and later of its neighboring societies in Asia, particularly Korea, Manchuria, China and Taiwan in the early 20th century. Many women in Korea, China and a few other Asian countries which were under Japanese colonial rule served as "comfort women", or sex slaves, and some had stillborn babies, but the cases of children born out of it living today remain little known. Among all these minoritized groups living in the Japanese archipelago, Koreans outnumbered the rest, culminating (from at one time over 2.2 million) to 2.4 million by 1944 (Lie 2004, p. 107).

The legacy of colonial rule under imperial Japan has led to the reality of "international marriages" being most prominent among those of Japanese and Korean nationality in the post-World War II years.[3] Since no apparent external physical features exist to distinguish people of Korean/Japanese heritage from ethnic Japanese, this difference in the intersectionality of the structural variable (colonialism) and race (skin color) has produced different patterns of the lived experiences of people of mixed heritage between the two regions.

Due to the invisibility of physical characteristics, as well as cultural and linguistic integration into the dominant Japanese culture, it is Korean nationality and names that mainly serve as important symbolic identity markers for many Korean–Japanese mixed people unlike during earlier decades when naturalization was challenging.[4] A husband's surname is usually kept while the wife loses her maiden name upon marriage, thus gender in mixed marriage patterns is a primary factor in shaping the (in)visibility of these markers.[5] While those individuals who pass as Japanese may live their lives carrying internal and

psychological struggles (Murphy-Shigematsu 2002),[6] an increasing number of young people of mixed heritage have opted to live beyond the dichotomy and proudly claim their Korean mixed heritage (Lee 2016).

The pattern of marriage or dating amongst the Ainu and the dominant Japanese has resulted from modern Japanese rule over indigenous peoples and internal colonialism and has continued domestically right up until contemporary Japan. Almost all Ainu—the indigenous group living primarily in Hokkaido, the northern island of Japan—are considered to be an ethnic Japanese mixture as a result of modern Japanese internal colonialism, which has continued domestically in contemporary Japan (Lewallen 2016; Sekiguchi 2016).[7] Even today, some women express their internalized aspiration to "erase Ainu blood" by outwardly marrying ethnic Japanese. Today there are a large number of Ainu descendants living in the Greater Tokyo Area as well.

### 4.3. The Presence of Foreign Military Forces and Bases since Occupation

Tens of thousands of African Americans in the US military, Africans (for example, from Senegal) in the French military and African-Caribbeans in the British military—overwhelmingly men—were stationed and/or fought in Europe during the second world war. Smaller numbers of each of these groups, but still in the thousands—again overwhelmingly men—were also present in those military forces during the first world war. As a result, a significant number of mixed-race children, with Black fathers and White mothers, were born in these nations (Campt 2004). Scholars have documented how the majority of these children remained in Europe with their White mothers after their fathers returned to these nations. It seems that the majority of them were abandoned. Several US military bases with significant numbers of African Americans remained in Europe at the end of World War II, though the numbers present were greatly diminished by the end of the 20th century. The numbers of Black military men were far higher for the US in Germany than in any other nation in Europe. Several scholars have looked closely at these mixed-race populations, with the most work being about Germany (Campt 2004; Aitken and Rosenhaft 2013). Continuing through the 1970s, these mixed populations were the primary focus of scholarship but, since then, the numbers of mixed people with African fathers has grown far more.

Under the GHQ occupation of Japan after World War II (1945–1952), many American GIs married, had sexual relationships with, or raped Japanese women Because of the phenotypical visibility of these children, the disappearance of biological fathers from their lives, the financial struggles of the women and the social stigma attached to the mothers, many of these children were abandoned (Ueda 2018). The most well-known orphanage for mixed-race children in the post-World War II period was the Elizabeth Saunders Home founded by a Japanese female philanthropist named Miki Sawada.

A similar situation has occurred up to the present in Okinawa, the island on whose military bases 70% of the U.S. forces in Japan are housed, as the result of agreements between the Japanese and U.S. governments. The concentration of U.S. military bases has produced a very noticeable population of mixed ancestry. Estimates of the numbers of children of mixed origins vary dramatically—from as little as 5000, to as many as 200,000. (Okamura 2017; Von Haas 2017). In addition, at present, in 2021, there are multiple U.S. bases still in Okinawa. The children of mixed Okinawan and (primarily White and Black) American ancestry have faced double discrimination—by both Okinawans and Japanese (Shimabuku 2016; Welty 2014). Children going to the Amerasian school that was built in 1998 for the bilingual and bicultural education of children of U.S. military personnel and Okinawan local women are called "Amerasian". Great inequality exists in power, language skills, resources and income between those with responsible American fathers and those raised by single Japanese mothers and between those with access inside the fence of the military base and not.

This particular pattern of interracial mixing demonstrates how the intersection of this structural factor with social variables of foreign fatherhood, such as race, gender and

others—in this case particularly nationality and language—can decisively shape social conditions and the lives of mixed-race people.

### 4.4. Work Migration

Millions of Europeans have moved across their continent, with the most common pattern through the 1980s being southern Europeans moving to the relatively richer areas of northern Europe, for example, from Italy, Spain, Portugal and Greece to England, Germany, Belgium and the Netherlands. Since the 1980s large numbers of people from Romania, as well as former Yugoslavia, relocated to Spain, France and the UK, as well as from Poland, Lithuania and Ukraine to Germany, the Netherlands and the UK. Scholars do not consider these populations to be racially mixed in the same way as, say, Whites and Blacks or Whites and Asians. However, these groups have been victims of significant racialized stereotypes and are often treated as if they are racially mixed in the nations to which they migrate. One of the major differences compared with earlier interracial mixing of Blacks and Whites is that given the phenotypical similarities, the mixed offspring of these populations are often indistinguishable from the populations with whom they are mixed. So, it is only other indicators (or signifiers)—such as names, religion and other cultural factors—that can be the basis of discrimination. In that regard, they are more similar to the mixing of Irish, Polish and Italians with the White populations of the United States.

In Asia, massive numbers of migrants, past and present, regarded one another as ethnically and religiously different, but not racially different, especially in the way race came to have significance in the Atlantic region. The racial significance of groups from outside Asia became common far later, and this pattern of racial/ethnic mixing has the longest history in Japan. In some metropolitan and pronouncedly multi-ethnic cities such as Nagasaki, Kobe and Yokohama, individuals came from such regions as Europe, North America, China and India, having immigrated to Japan as early as the beginning of the Meiji period, or even earlier in small numbers in Nagasaki.[8] They came as missionaries, merchants, teachers and individual travelers. These cities are known to be more cosmopolitan and tolerant to diversity for this reason. However, these early immigrants mostly consisted of Europeans or European Americans, Chinese or Indians, not Africans or African descendants or people of other regions, thus limiting the kind and degree of its "cosmopolitanism".

The aspiration for Westernization, which was then conflated with civilization, started in the beginning of the Meiji period (1868–1912) soon after Japan started to engage more actively with foreign countries. Although the meanings associated with mixed White people have shown significant and complex changes over the century and a half since the Meiji Restoration in 1868, particularly when intertwined with Japanese nationalism, imperialism and racial ideology, the racial hierarchy they imported from European and American "science" has fundamentally remained little changed and thus has affected the gaze toward and representations of people of mixed race/ethnicity.

The significance of migration and travel has been increasing due to contemporary globalization, which has been bringing more diversity in skin color, "race", class, social status, nationality and religion in today's Japanese society. A key dimension of recent decades in Japan is the arrival of thousands of families of Japanese ancestry from Brazil and Peru. The majority of these families included non-Japanese spouses, so the children are racially mixed (Nishida 2018).

Again, the intersectionality with social variables has often determined social position or representation of those with mixed race. As discussed in the beginning, there is a high visibility of and relative "admiration" for White mixed race in the media and society at large. Some, such as children of Filipina women and Japanese men, or Brazilian-mixed Japanese from Brazil, stand as racially ambiguous. Although many Japanese Filipina children, often abbreviated as JFC, have mothers who used to work as entertainers before meeting their Japanese husbands, the increasing number of Filipina nurses and caregivers and Filipinas in other skilled occupations are rapidly replacing the stereotypical images associated with Filipina women as entertainers (Parrenas 2011).

For example, by 2013, there were 146,000 Filipina spouses of Japanese men living in Japan (Suzuki 2017, p. 127, fn 7). A distinctive feature of Japan is that, whereas across southeast Asia women commonly traveled to work as domestics, nurses, caregivers and other types of workers, Japan was, from the 1970s to the mid-2000s, the most important destination in the world for women entertainers in nightclubs and similar places (Suzuki 2017). Most of these women were socially upwardly mobile, but many other women experienced downward mobility while in Japan, especially if they ended up divorced or widowed.

Brazilian-mixed Japanese from Brazil stand as ambiguous not only racially but also social-status wise. As if mixed with Whites, they may be included in the Western-Japanese mixed race but, on the other hand, Japanese–Brazilian mixed people or those who have Japanese Brazilian parents may be looked down upon as semi-skilled or unskilled laborers who come in on a special work-permit category granted to second and third generation Japanese descendants in South America. Japanese–Brazilian young women perform as hāfu moderu (half models) by hiding or toning down their Brazilian Japaneseness, as it is associated with migrant laborers or the lower socio-economic class, instead accentuating their Brazilianness for upward socio-economic mobility (Watarai 2014). In these cases, gender and class/social status have played significant roles in their social position and representations in Japan.

It is not unusual for first-generation immigrant workers dedicated to unskilled labor or entertainment to be regarded as being in a different class from those who came individually to Japan as skilled workers in business, religion, or education.

*4.5. Marriage Migration*

There are small but significant numbers of women from various nations of Southeast Asia, the Philippines, the Caribbean and Brazil, who have become permanent residents of Western Europe, especially France, the UK, Spain and Italy (Parrenas 2015). Many arrived in Europe to take up positions in the so-called caring professions—as home aides and nannies, working in hospitals, hotels, catering and prostitution (Andall 1992). Some women from Southeast Asia fell into the category of "mail-order brides", while many others have had arranged marriages through more modern agencies (Charsley 2012). Some women from Brazil and the Caribbean, especially the Dominican Republic, became the wives of men who visited those nations for work or leisure or prostitution (Brennan 2004). In terms of interracial marriage and children, these groups have not received much attention in scholarship. Since the fall of the Berlin Wall, far greater numbers of women from Eastern Europe have become brides of men in Western Europe. The emphasis with those relationships is on cultural differences rather than racial differences. In addition, their children are phenotypically very similar to the White populations of Western Europe.

In Japan, where population decrease, especially in rural areas, has become a serious social problem, men engaged in farming, fishing and other rural industries have difficulty finding wives. In increasing numbers, so-called "Brides" immigrants have come to marry Japanese men to start families (Constable 2005; Piper 2013).[9] This kind of international marriage is clearly race and gender intersected. Many wives are from Korea, China and the Philippines. In the mid 1890s, some local governments in rural areas, though few, organized a tour for their local men to arrange meetings with women in East or Southeast Asia, although the practice has stopped. The Tozawa Village in Yamagata Prefecture has earned a reputation as a successful model for bringing in foreign wives as permanent residents (Ando 2009). In Japan at the start of the 21st century, Chinese women were the most likely foreigners to marry Japanese men and live in Japan, followed by Filipinas. Divorce rates of non-Japanese women and Japanese men are high, however. The rates were around 40% by the late 1990s, and much higher in the 21st century (Suzuki 2017). A small but increasing number of men from Europe as well as from other regions of the world are married to Japanese women and live in Japan (Debnár 2016).

When these structural factors intersect with the following variables, each of which has particular meaning and value in each social context, the experiences and representations of individuals born under those circumstances of marriage are shaped, along with societal values/aesthetics and racial hierarchical ideology. In the next section, we identify how each of the above six structural factors have operated in Japan when intersected with the four major social variables (a–d), presenting examples of the intersectionalities.

*4.6. Others*

It is highly likely that there are other factors at work, shaping interracial dating and people of mixed racial descent. In Europe, there are also hundreds of thousands of students under the ERASMUS program, studying for periods of months or longer in foreign nations, and there must be some mixed children resulting from this population movement. ERASMUS is a European exchange program, but, as we note, not all Europeans are White.

**5. Discussion**

In this article we have highlighted some of the limitations of the analysis of variables in the Atlantic region and highlighted key variables in the Pacific that require greater attention. By juxtaposing the Atlantic and Japan, we have highlighted an important set of variables that are configured very differently in the Pacific than in the Atlantic. This includes six relevant structural features and four variables. The six features are: (1) slavery; (2) other types of conquest and colonialism/imperialism; (3) the presence of foreign military forces and bases since the occupation; (4) work migration; (5) marriage migration; (6) others. These features intersect with four variables: (a) race: a-1: skin color and societal notions of racial categories, a-2: the ideology of "bloodlines"; (b) gender and sexuality; (c) class and societal status; (d) other societal and cultural differences (nationality, language, religion, etc.).

The limited nature of the existing literature raises several issues. The first issue is the limited relevance of the most prominent Atlantic region features and legacy for the Pacific region, along with the decreasing relevance of this legacy for the Atlantic region itself. These two issues are implied strongly by the growth of super-diversity. We have identified some of the major ways in which the Pacific manifests different dynamics—or at least different configurations of similar dynamics—than the Atlantic, as revealed in the six points we identified. At present, the Pacific lacks any association with slavery in the sense equivalent to the one in the Atlantic, has greater association with other types of conquest, with work and marriage migration and with the presence of foreign military. These factors continue to play important roles.

On both sides of the Atlantic, the legacy of slavery and the Black/White binary will become far less important, given the rise in super-diversity. This is already the case for people of mixed racial descent in California, where Latinos and Asians outnumber Black people. In addition, it is the case in many cities in Western Europe with mixed populations, including London, Paris, Amsterdam, Berlin, Rome and elsewhere. In addition, among Black people on both sides of the Atlantic, we have already mentioned that mixed-race people with one Black parent are increasingly likely to have an African parent, rather than one descended from the enslaved populations of the Americas.

This is relevant because Black people of mixed racial descent in the Atlantic emerged largely in contexts regarded as illicit and/or illegitimate; that is, born under slavery, outside marriage and as the victims of vicious racist stereotypes of ability and sexuality. These relationships overwhelmingly involved White men dominating Black women, often involving limited choices by women and frequently rape. The children were consistently stigmatized. At present, across both regions, there is increasing choice in interracial relationships, far less stigma attached and fewer associations of illegitimacy attached to the children. The super-diversity that has emerged has led to far more complex patterns of interracial dating, marriage and people of mixed racial descent than ever before. These relationships have brought a far more diverse—and divergent—set of connotations and meanings. This is a positive development. As these patterns unfold, we suspect that

ethnicity—in the form of religion, language and maybe other issues—will become more prominent in marriages and identities.

Attention to gender continues to be important in both regions. Interracial marriages between U.S. soldiers from U.S. military bases and local women, especially in Asia, and marriage migration patterns demonstrate how the intersectionality of gender, race and state have affected whether or not the children of mixed heritage have privileges or disadvantages.

In the patterns that continue to unfold, people of mixed racial descent in both regions will play far bigger roles than they have in the past (as will the parents involved in these relationships). Among them are emerging scholars with mixed-race backgrounds who are arising both in Japan and other regions in the Pacific. People of mixed race in both regions are actively involved in scholarship, literature and filmmaking, as well as among musicians, artists, poets and celebrities. They are also involved in a wide range of highly impactful social media. Many more are speaking publicly, not just superstars and celebrities in sports and television, but also a much wider cross-section. This will lead to more balanced and informed discussion and debate on these issues—which will in turn yield a more extensive analysis, especially in their debates with mono-racial people. This particular pattern is already far more developed in the Atlantic than it is in Japan, with groups and organizations that advocate for people of mixed racial descent, some of which have existed for decades. However, mixed-race identities are already evident and growing in Japan and elsewhere, and demographics suggest that they will continue to grow for the foreseeable future. There is some exchange between the two regions, with positive effect, but the impact is negligible where English is not spoken or used, given the dominance of Anglophone media (King-O'Riain et al. 2014). Japan is a case in point because the number of people of mixed race is relatively miniscule and the impact of ideas from across the globe is limited by language barriers, although mixed-roots organizations in Japan have benefited from several significant exchanges with groups and organizations in the United States, such as Hapa Japan and the Mixed and Remixed Festival.

In these various developments, the state in each nation has played a decisive and powerful role historically and continues to play a powerful role at present. This involves both direct actions (laws, citizenship and marriage policies) and indirect actions (work and marriage migration and trans-racial adoption) that affect these patterns. One major issue will be the extent to which people of mixed racial descent mobilize to change state policy, for example, regarding the census, or the rights of foreign spouses for citizenship.

At the present time, it is clear that while the emerging scholarship on these issues in Japan (and the Pacific) is deeply grounded in the literature of scholars in the Atlantic, the reverse is far from true (Rocha 2016; Rocha and Fozdar 2017). In fact, it seems that scholars of the Atlantic do not generally take into account the key variables or analysis of the Pacific. In addition, if they are, then they do not cite the relevant literature.

In order to broaden the scope of the literature in mixed race studies we can start by juxtaposing the circumstances of the Pacific with those of the Atlantic. Such a model needs to take account of the myriad ways in which mixed-race identities have developed and are developing without the legacy or stigma of slavery and its legacies that continue to be a signifier of such relationships in the Atlantic region. It will need to take account of the far greater salience of work, marriage and migration. In addition, it will need to take account of the continued presence of overseas-stationed military personnel.

Needless to say, given the recent emergence and limited number of groups—especially groups of mixed-race people—addressing mixed race issues, it will take more time for them to have an impact on national attitudes and beliefs. Examples of mobilization by groups of mixed-race people—and by their parents—is evidenced in several nations across the Atlantic, and they have already had a significant impact on state policy and census format in the U.S. and the UK (Campion 2017; Song 2017). We do not have significant evidence of such actions in Japan, though there are signs that it is beginning. In addition, signs for the rights of nationals of Japan to bring their spouses to that nation are also being raised (Takezawa 2016). Increased consideration of the role of the military is also likely to

refocus attention on the significant, if largely neglected, role—and its legacies—that the U.S. military played in Europe, especially in Germany, the UK and Italy. In addition, there has been very little discussion of the role of Africans fighting for France stationed in Germany during World War II (Aitken 2016; Aitken and Rosenhaft 2013).

## 6. Conclusions

In the existent literature on the theories of mixed race, while some writers claim to provide a global model, the model is, in reality, based almost exclusively on trans-Atlantic experiences and may be called more appropriately the Atlantic model. The structural and institutional dynamics of racialization, along with the ideological belief systems and cultural practices, the differential significance attached to ancestry, blood, color and race in Japan reveal fundamentally different characteristics and consequences to those in the Atlantic region. The alternative model we have proposed, consisting of six features and four variables, we hope, will contribute to enriching our understanding beyond the Atlantic, in this case Japan, as an example of the Pacific.

**Author Contributions:** Conceptualization, Y.T. and S.S.; methodology, Y.T. and S.S.; software, Y.T. and S.S.; validation, Y.T. and S.S.; formal analysis, Y.T. and S.S.; investigation, Y.T. and S.S.; resources, Y.T. and S.S.; data curation, Y.T. and S.S.; writing—original draft preparation, Y.T. and S.S.; writing—review and editing, Y.T. and S.S.; visualization, Y.T. and S.S.; supervision, Y.T. and S.S.; project administration, Y.T. and S.S.; funding acquisition, Y.T. All authors have read and agreed to the published version of the manuscript.

**Funding:** This research was funded by the Japan Society for the Promotion of Science, Grant-in-Aid for Scientific Research (S) KAKENHI grant [16H06320] "Integrated Research into the Processes and Mechanisms of Racialization".

**Acknowledgments:** The authors acknowledge the valuable comments on this article received by Rebecca Chioko-King O'Riain, Miri Song, Duncan Williams, Remi Joseph-Salisbury, and Lyle De Souza.

**Conflicts of Interest:** The authors declare no conflict of interest.

## Notes

[1]    No numerical data are available for those who identify themselves as mixed race/ethnicity.

[2]    For a discussion of internal colonialism in contrast with external colonialism, see, for example, (Blauner 1969).

[3]    Emperor Akihito (then) made a memorable comment at a press conference in 2001, one year before the World Cup hosted by Japan and the Republic of Korea, "I, on my part, feel a certain kinship with Korea, given the fact that it is recorded in the *Shoku Nihongi* (Chronicles of Japan, compiled in 797), that the mother of Emperor Kammu (reign 781~806) was of the line of King Muryong (reign 501~523) of the Kingdom of Paekche". This comment, admitting the so-called mixed blood even in the Imperial lineage, was his implicit message to Japanese people, especially some ultra-nationalists and anti-Korean racists, in the hope of having a good neighboring relationship with South Korea.

[4]    Koreans living in Japan were deprived of the Japanese nationality by the San Francisco Peace Treaty of 1952, and their nationality was first marked "Korean", and later, those who chose the affiliation with South Korea were given South Korean citizenship by the Republic of South Korea, while the rest remained "Korean".

[5]    Even though historically ethnic Koreans faced obstacles in obtaining naturalization, especially before the 1990 change of immigration law, today the situation has changed in the other direction. Nationalities and surnames serve as important ethnic markers for those who opt to maintain and embrace them.

[6]    The story of Korea/Japanese Fusae (pseudonym) in a documentary film, "Hafu" also demonstrates this point. http://hafufilm.com/en/ (accessed on 3 December 2020).

[7]    Marriage and assimilation between the Okinawans (*Uchinanchu*) and mainland Japanese (*Yamatonchu*) apply here as well. The United Nations Committee on the Elimination of Racial Discrimination names the Ainu, *hisabetsu buraku*, and Okinawan people as the historical minorities of Japan.

[8]    See (Leupp 2017) for a rare historical analysis of the relationships between western men and Japanese women from the 16th century to the turn of the 20th century.

[9]    Men in rural areas in Korea are in a similar situation and welcoming migrant wives. See Ji-Hyun Ahn's book *Mixed-Race Politics and Neoliberal Multiculturalism in South Korean Media*.

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
