# Peer review of "Theorizing People of Mixed Race in the Pacific and the Atlantic"

_socsci, doi:10.3390/socsci11030124_

Round 1

Reviewer 1 Report

This is a nicely written and thought-provoking paper that would certainly generate interest and debate amongst the readership of Social Sciences.

I was not altogether persuaded by the approach. The authors open with the statement : ‘The most extensive theoretic and empirical studies of people of mixed racial descent extant today have addressed nations across the Atlantic’. There has been a huge increase in scholarly work on mixing and mixedness in the Pacific and other world regions in the last decade, little of which is cited. For example, The Palgrave International Handbook of Mixed Racial and Ethnic Classifications  (Rocha and Aspinall 2020) (not cited) has 12 chapters on Asian and the Pacific.

This substantial body of scholarly work has done much to redress the imbalance in mixed race scholarship that had hitherto existed with respect to global coverage and focused on the specific mechanisms of mixing and mixedness in diverse national contexts.

Nor was I persuaded of the merits of a search for ‘a truly global approach for theorizing people of mixed racial descent’. Mixing and mixedness are shaped by complex processes of ethnogenesis within particular national contexts which in turn lead to a diversity of processes of racialisation. I’m not sure how useful it is to talk of ‘Atlantic models’ and ‘Pacific models’ and question the analytical and explanatory value that these capacious regional approaches can confer. There clearly is scope for such a focus with respect to particular issues (as the work of Morning and Rallu has demonstrated) but a ‘theorizing of people of mixed race’ (at this level of generality) does seem challenging.

Yet the authors have set down this challenge and this, in itself, is interesting and in my view merits publication.

Minor issues: p. 3 of abstract. Of mixed racial is limited (descent missing)

Author Response

We appreciate the review and have taken them into account in the revised version. We do in fact already cite the handbook by Rocha and Aspinall, as well as several other publications on the Pacific. We maintain that despite an increase in writings on the Pacific, they do not begin to match the volume or scope of writings on Atlantic nations so our contention stands. And we have modified the idea of 'a truly global approach' and offer instead a more encompassing approach.  

Reviewer 2 Report

A great paper. Interesting and important. Needs a few clarifications on some points.

Author Response

We thank the reviewer for the feedback. We have made the minor changes and spellcheck. And we have clarified some of the points, in particular the idea of a truly global model. Overall we feel that articles maintains its focus, clarity and coherence.